# On Transformation Form-Invariance in Thermal Convection

**DOI:** 10.3390/ma16010376

**Published:** 2022-12-30

**Authors:** Gaole Dai, Jun Wang

**Affiliations:** 1School of Sciences, Nantong University, Nantong 226019, China; 2School of Physics, East China University of Science and Technology, Shanghai 200237, China; 3School of Mathematics, East China University of Science and Technology, Shanghai 200237, China; 4Wenzhou Institute, University of Chinese Academy of Sciences, Wenzhou 325001, China

**Keywords:** transformation optics, thermal convection, invisibility, thermal metamaterials

## Abstract

Over the past two decades, effective control of physical fields, such as light fields or acoustics fields, has greatly benefited from transforming media. One of these rapidly growing research areas is transformation thermotics, especially embodied in the thermal conductive and radiative modes. On the other hand, transformation media in thermal convection has seldom been studied due to the complicated governing equations involving both fluid motion and heat transfer terms. The difficulty lies in the robustness of form invariance in the Navier–Stokes equations or their simplified forms under coordinate transformations, which determines whether the transformation operations can be executed on thermal convection to simultaneously regulate the flow and thermal fields. In this work, we show that thermal convection in two-dimensional Hele–Shaw cells keeps form-invariance, while its counterpart in general creeping flows or general laminar flows does not. This conclusion is numerically verified by checking the performances of invisible devices made of transformation media in convective environments. We further exploit multilayered structures constituted of isotropic homogeneous natural materials to realize the anisotropic inhomogeneous properties required for transformation media. Our results clarify the long-term confusion about the validation of the transformation method in thermal convection and provide a rigorous foundation and classical paradigm on inspiring various fascinating metadevices in both thermal and flow fields.

## 1. Introduction

Transformation media, designed by transformation optics (TO) [1,2], can realize exotic functions such as invisibility in different scenarios from wave systems to diffusion processes [3,4]. Requiring the form invariance of governing equations under a coordinate transformation induced by a homomorphism map from the original space (virtual space) to the real world (physical space) [4], TO can predict the transformed material parameters to map the physical field correspondingly. An important branch of transformation media is the thermal metamaterials, which can help us regulate heat transfer at will against the backdrop of the energy crisis and the increased demand for thermal management [5,6]. As we know, heat transfer includes three basic modes: conduction, radiation and convection. Effective manipulation of heat transfer requires consideration of all three modes. The study of thermal metamaterials began with the establishment of transformation thermotics [7,8], a diffusion analogue of TO in thermal conduction. Thermal metamaterials in radiation, on the other hand, can be classified into electromagnetic metamaterials in principle. Finally, applying TO on thermal convection seems more difficult. Thermal convection is a coupled process of heat and mass transfer, and its governing equations are quite complicated. We must consider the heat transfer equation (advection–conduction equation) and the hydrodynamic equations (the law of continuity and the Navier–Stokes (NS) equations) at the same time. It has been proved that the heat transfer equation and the law of continuity for fluid motion both meet the requirement of TO [9,10]. Then, applying TO for general thermal convection depends on whether the NS equations are form-invariant. However, even considering only isothermal flows, there is currently a lack of a consistent conclusion on this question. Conflicting viewpoints appear in different literature [6,10,11,12,13,14].

In this work, we study the convective heat transfer, especially in pipe flows with a small Reynolds number, i.e., creeping flows or Stokes flows. We show that TO-based invisible devices to regulate heat transfer can only work perfectly in the two-dimensional (2D) Stokes flow in slab, saying Hele–Shaw flow. While in the general NS-based thermal convection or three-dimensional Stokes-based thermal convection, the conditions for applying TO are not strictly satisfied. For verifying this theoretical conclusion, we construct some counter-examples in which the devices lose thermal invisibility and other desirable functions such as cloaking, concentrating and rotating the thermal fields. In other words, we give a direct and valid proof that thermal convection governed by the Stokes equation (a simplified form of the NS equations) or the NS equations themselves is not form-invariant under coordinate transformations. In addition, for 2D Hele–Shaw flows, simplified designs of the invisible devices are given using the multilayered structures made of isotropic homogeneous materials, demonstrating the ability to simultaneously manipulate thermal and flow fields.

## 2. Theoretical Modeling

First, we introduce the governing equations of heat convection in the virtual space, which is usually isotropic. For simplicity, we only consider incompressible Newtonian fluid and steady creeping flow in this work. The thermodynamic state of general non-isothermal flows can be described by (T,p,v), which can be determined by [15]
(1a)$$\begin{array}{c} \nabla \cdot \left(-\kappa \nabla T+\rho CTv\right)=0, \end{array}$$
(1b)$$\begin{array}{c} \nabla \cdot \left(\rho v\right)=0, \end{array}$$
(1c)$$\begin{array}{c} \nabla p=\nabla \cdot \left(\mu \nabla v\right). \end{array}$$

Equation ([Disp-formula FD1a-materials-16-00376]) is the advection–conduction equation for heat transfer, where κ is the thermal conductivity, *T* is the temperature, ρ is the density, *C* is the heat capacity, and v is the velocity. Equation ([Disp-formula FD1b-materials-16-00376]) is the law of continuity for fluid flow. Equation ([Disp-formula FD1a-materials-16-00376]) is the simplified Navier–Stokes equations (the Stokes equation), where *p* is the modified pressure or the pressure perturbation relative to hydrostatic pressure [16] (we do not consider gravity-driven flows so the hydrostatic pressure term and the gravity can be balanced and offset in Equation ([Disp-formula FD1c-materials-16-00376])) and μ is the shear viscosity. For general laminar flows or even turbulence, we need to add the inertial term ρ(v·∇)v back into the right side of Equation ([Disp-formula FD1c-materials-16-00376]) and re-obtain the NS equations. If TO cannot be applicable to the thermal convection in some creeping flow, then the more general cases describing the NS equations cannot meet the requirement of TO either.

Previous studies have proved that both Equation (1a,b) are form-invariant under a homomorphism map (geometric transformation) from the position vector r in the virtual space to that in the physical space (denoted by $r^{\prime }$), i.e., $r\mapsto r^{\prime }$. Their forms in the physical space should be (the quantities and operators are superscripted for distinction)
(2a)$$\begin{array}{c} \nabla^{\prime }\cdot \left(-\kappa^{\prime }\cdot \nabla T^{\prime }+\rho^{\prime }C^{\prime }T^{\prime }v^{\prime }\right)=0, \end{array}$$
(2b)$$\begin{array}{c} \nabla^{\prime }\cdot \left(\rho^{\prime }v^{\prime }\right)=0, \end{array}$$
where the temperature and velocity in the physical space become [9,10]
(3a)$$\begin{array}{c} T^{\prime }\left(r^{\prime }\right)=T\left(r\left(r^{\prime }\right)\right), \end{array}$$
(3b)$$\begin{array}{c} v^{\prime }\left(r^{\prime }\right)=\frac{J\left(r^{\prime }\right)v\left(r\left(r^{\prime }\right)\right)}{\det J(r^{\prime })}, \end{array}$$
and the material properties satisfy [8,9,10]
(4a)$$\begin{array}{c} \kappa^{\prime }\left(r^{\prime }\right)=JJ^{⊤}\det J^{-1}\kappa \left(r\left(r^{\prime }\right)\right), \end{array}$$
(4b)$$\begin{array}{c} \rho^{\prime }\left(r^{\prime }\right)=\rho \left(r\left(r^{\prime }\right)\right), \end{array}$$
(4c)$$\begin{array}{c} C^{\prime }\left(r^{\prime }\right)=C\left(r\left(r^{\prime }\right)\right). \end{array}$$

Here, $J(r^{\prime })$ or J is the Jacobian $\partial r^{\prime }/\partial r$. $J^{⊤}$ and detJ are its transpose and determinant, respectively. In particular, the thermal conductivity $\kappa^{\prime }$ is now a rank-2 tensor. The key in the above transformation rules is how to realize the velocity distribution [Equation ([Disp-formula FD3b-materials-16-00376])]. In this work, we assume the density and heat capacity are both constants, so Equation (4b,c) indicate the density and heat capacity are not changed after the transformation. The uniform density (at least along the direction of gravity or the *z* axis) also rules out the possibility of gravity-driven flows.

Further, it is easy to find that with Equation ([Disp-formula FD1b-materials-16-00376]), Equation ([Disp-formula FD1c-materials-16-00376]) can be reduced to a Laplacian equation for the pressure [17]:(5)$$\frac{1}{\mu }\nabla \cdot \nabla p\left(r\right)=\nabla \cdot \left(\frac{1}{\mu }\nabla p\left(r\right)\right)=0,$$
if μ and ρ are both constants (and the flow is irrotational). Equation (Equation 5) has the same Laplacian form as Darcy’s law for creeping flows in porous media, which is form-invariant under coordinate transformations [18]. From a mathematical point of view alone, Equation (Equation 5) is also form-invariant, and its form in the physical space should be (see detailed derivations in Appendix A)
(6)$$\nabla^{\prime }\cdot \left(JJ^{⊤}\det J^{-1}\frac{1}{\mu }\nabla^{\prime }p^{\prime }\left(r^{\prime }\right)\right)=0.$$

In particular, the pressure in the physical space satisfies
(7)$$p^{\prime }\left(r^{\prime }\right)=p\left(r\left(r^{\prime }\right)\right).$$

So, the transformation rule for the shear viscosity can be obtained:(8)$$\mu^{\prime }\left(r^{\prime }\right)=J^{-⊤}J^{-1}\mu \det J.$$

The shear viscosity in the physical space is now a rank-2 tensor. This is the theoretical basis for some previous works to design transformation hydrodynamic or convective metamaterials in Stokes flows [11,12,19], although their derivation processes might differ. However, the corresponding anisotropic form of Equation ([Disp-formula FD1c-materials-16-00376]) in the physical space will be controversial. It must be [11,12,19]
(9)$$\nabla^{\prime }p^{\prime }\left(r^{\prime }\right)=\nabla^{\prime }\cdot \left(\mu^{\prime }\left(r^{\prime }\right)\cdot \nabla^{\prime }v^{\prime }\left(r^{\prime }\right)\right),$$
but it is impossible to obtain Equation (Equation 6) from Equations ([Disp-formula FD2b-materials-16-00376]) and (Equation 9) because $\mu^{\prime }$ is spatially varying. Equation (Equation 6) can only be considered an approximation if $JJ^{⊤}\det J^{-1}$ causes only small spatial inhomogeneities. In fact, one approach to Equation (Equation 6) is just using the (generalized) Darcy’s law in the physical space by filling porous media [10,18,20]. Another feasible approach is using the Hele–Shaw flow [21,22].

In this work, we use Hele–Shaw flow to refer to a kind of 2D or planar flow. It is an approximation to a three-dimensional (3D) model called the Hele–Shaw cell [21,22], i.e., the pipe flow with a rectangular section [Figure 1a]. The depth of the cell, denoted by *h*, is much smaller than the width and the length, which have the same value *L* in our model. Using the lubrication approximation, the planar average velocity (integrating along the *z*-axis) is [23]
(10)$$\bar{v}\left(x,y\right)=-\frac{h^{2}}{12\mu }\nabla p.$$

If we use an approximation $v\left(x,y\right)=\bar{v}\left(x,y\right)$, the 3D Hele–Shaw cell can be viewed as a 2D model [Figure 1b]. Substituting the NS equations with the Hele–Shaw equation [Equation (Equation 10)] in Equation (1a–c), we can obtain the governing equations for thermal convection in a Hele–Shaw cell. Equation (Equation 10) has the same form as Darcy’s law, with an effective permeability as $\frac{h^{2}}{12\mu }$. Combining Equations ([Disp-formula FD1b-materials-16-00376]) and (Equation 10), we have
(11)$$\nabla \cdot (\frac{\rho h^{2}}{12\mu }\nabla p)=0,$$
which reduces to Equation (Equation 5) when *h* and ρ are both constants. If *h* is an invariant constant under coordinate transformations, the generalized anisotropic form of Equation (Equation 10) in the physical space is [14]
(12)$$\mu^{\prime }\cdot v^{\prime }\left(x^{\prime },y^{\prime }\right)=-\frac{h^{2}}{12}\nabla^{\prime }p^{\prime },$$
and the corresponding form of Equation (Equation 6) can be obtained from Equations (Equation 12) and ([Disp-formula FD2b-materials-16-00376]):(13)$$\nabla^{\prime }\cdot \left(JJ^{⊤}\det J^{-1}\frac{\rho h^{2}}{\mu }\nabla^{\prime }p^{\prime }\right)=0.$$

In other words, the Hele–Shaw flow itself is form-invariant if there exists a rank-2 viscosity tensor that satisfies Equation (Equation 8) [14]. Although we consider a constant density here, we can see a spatially varying density in Equation (Equation 13) shall not violate the form invariance as long as it satisfies Equation ([Disp-formula FD4b-materials-16-00376]) and does not cause vertical density gradient that results in an extra natural convection. It should be pointed out that this anisotropic Hele–Shaw model [Equations (Equation 12) and (Equation 13)] can indeed be derived from the anisotropic NS equations in shallow geometry [14,24]. So, 2D thermal convection described by Equations (1a,b) and (Equation 10) meet the requirement of TO, share the same forms as Equations (2a,b) and (Equation 12) and can be derived from realistic 3D models under the lubrication approximation.

## 3. Numerical Verification

Here, we perform a set of numerical simulations to confirm our different conclusions about the form invariance of thermal convection in general creeping flows versus special Hele–Shaw flows. Naturally, we would expect functional devices composed of transformation media to work in Hele–Shaw flows and fail in some creeping flows. We use a trick to achieve these two purposes at the same time through simulations for a set of pipe flows with different geometric parameters. As we said in Ref. [14], if Equations ([Disp-formula FD1c-materials-16-00376]) and (Equation 10) are both form-invariant, so is their combination, the modified Hele–Shaw equation with the viscous stress term:(14)$$\nabla p-\nabla \cdot \left(\mu \nabla v\right)+\frac{12\mu }{h^{2}}v=0.$$

Its form in the physical space is
(15)$$\nabla^{\prime }p^{\prime }-\nabla^{\prime }\cdot \left(\mu^{\prime }\cdot \nabla^{\prime }v^{\prime }\right)+\frac{12}{h^{2}}\mu^{\prime }\cdot v^{\prime }=0,$$
without changing any transformation rules. When *h* is small enough, Equations (Equation 14) and (Equation 15) approximate Equations (Equation 10) and (Equation 12), respectively. We will compare the performance of transformation media when replacing the NS equations in the virtual space with Equations (Equation 14) and (Equation 10). If the simulation results show that Equation (Equation 10) is form-invariant but Equation (Equation 14) is not, then the only reasonable explanation is that Equation ([Disp-formula FD1c-materials-16-00376]) changes its form under coordinate transformations.

Three invisible devices are used in the following simulations: the cloak, the concentrator and the rotator. The transformation (written with polar coordinates r=(r,θ) and $r^{\prime }=\left(r^{\prime },\theta^{\prime }\right)$) corresponding to a cloak [Figure 1c,d] is [1]
(16)$$\left\{\begin{array}{c} r^{\prime }=R_{1}+\frac{R_{2}-R_{1}}{R_{2}}r,\;0<r<R_{1},\\ \theta^{\prime }=\theta . \end{array}\right.$$

The cloak itself is an annular shell ($R_{1}<r^{\prime }<R_{2}$). The transformation isolates the area $0<r^{\prime }<R_{1}$, which the heat flux and fluid flow cannot enter in. The area outside the cloak undergoes an identity transformation, avoiding being disturbed by the cloak and the object inside it. In other words, the cloak is invisible if we only observe the outside environment. The transformation for the concentrator [Figure 1c,e] can be expressed as [25]
(17)$$\left\{\begin{array}{c} r^{\prime }=\frac{R_{1}}{R_{m}}r,\;r<R_{m},\\ r^{\prime }=\frac{R_{1}-R_{m}}{R_{2}-R_{m}}R_{2}+\frac{R_{2}-R_{1}}{R_{2}-R_{m}}r,\;R_{m}<r<R_{2},,\\ \theta^{\prime }=\theta , \end{array}\right.$$
where $R_{m}$$\left(R_{1}<R_{m}<r<R_{2}\right)$ is an auxiliary parameter that determines the concentrating effect inside the concentrator. The transformation for the rotator [Figure 1c,f] is [26]:(18)$$\left\{\begin{array}{c} \theta^{{}^{\prime }}=\theta +\theta_{0},\;r<R_{1},\\ \theta^{{}^{\prime }}=\frac{\left(r-R_{2}\right)}{R_{1}-R_{2}}\theta_{0}+\theta ,\;R_{1}<r<R_{2},\\ r^{\prime }=r, \end{array}\right.$$
where θ is the angle by which the fields inside the rotator are expected to be rotated counterclockwise. Like the cloak, the concentrator and the rotator are also invisible shell devices occupying the same domain. In simulations, the geometric parameters of the 2D flow include $L=D=10^{-2}$ m, $R_{1}=2\times 10^{-3}$ m and $R_{2}=3\times 10^{-3}$ m. The values of $R_{m}$ and θ are $2.9\times 10^{-4}$ m and −60°, respectively, for the concentrator and the rotator. A heat source with a temperature of 298 K is placed on the left side, while another with a temperature of 293 K is placed on the right, so the thermal bias is 5 K. A pressure difference Δp is applied on the *x* direction. The left side is the inlet of flows, and the right side is the outlet. The material in the virtual space is referenced to water. Its property parameters include the thermal conductivity κ=0.6 W m${}^{-1}$ K${}^{-1}$, the viscosity $\mu =10^{-3}$ Pa s, the density ρ=1000 kg m${}^{-3}$ and the heat capacity C=5000 J kg${}^{-1}$ K${}^{-1}$. For simplicity, we do not consider the thermal response of density or viscosity here. Such ‘nonlinearity’ does not affect the validation of TO [20] but will make the details of material transformation much more complicated. In particular, the area inside the cloak is set to solid, so the velocity at $r^{\prime }=R_{1}$ is zero. In addition, referring to the simulation approach in Ref. [27], thermal insulation condition is also applied at $r^{\prime }=R_{1}$. In this way, the solid domain can be removed from the geometry in the simulation. This treatment makes sense because, according to Equation (Equation 16), we do not actually know the distribution of the fields inside the cloak.

First, we give the simulated temperature distributions in Figure 2. Equations (Equation 14) and (Equation 15) are used in the first three column plots which have different values of h/L; Δp also takes different values for corresponding h/L to make sure the Reynolds numbers are similar in all the cases; h/L in the first two columns is 0.001 and 0.01, and the corresponding pressure difference is 50 Pa and 0.5 Pa, respectively. The flows can be approximated as Hele–Shaw flows. Comparing the contour plots and isotherms of three devices and the virtual space, we find that the cloaking, concentrating and rotating functions inside the devices are all realized. In addition, the three devices are also indeed invisible to the background. Of course, if we observe carefully, we will find that the isotherm in Figure 2(a2) deviates slightly from Figure 2(b2) because the no-slip condition set at $r^{\prime }=R_{1}$ and the required velocity transformation rule are not completely consistent. In addition, the extreme parameters of the cloak here, with values of zero or infinity, cannot be accurately achieved in numerical simulations. The third column, with regard to h/L=0.1 and pressure difference equal to 0.005 K, demonstrates a quite obvious failure of the invisible devices, especially the cloak [Figure 2(a3)] and the rotator [Figure 2(d3)]. The concentrator [Figure 2(c3)] shows the smallest deviation, although we have used a very large $R_{m}$. This is because the spatial inhomogeneity and anisotropy generated by the concentrating transformation [Equation (Equation 17)] is much smaller than the other two devices. From the first three columns, it can be asserted that Equations (Equation 14) and (Equation 15) do not satisfy the requirements of TO. The last column, also taking h/L=0.1, is based on Equations (Equation 10) and (Equation 12). Although the 3D pipes corresponding to these flows actually cannot be called Hele–Shaw cells due to the value of h/L, here we focus on checking the form-invariance. The contour plots and isotherms in the last column are similar to those in the first two columns, and the three devices work well. So, the form-invariance of thermal convection in Hele–Shaw flows can be confirmed. In contrast, general thermal convection is not form-invariant when the lubrication approximation is not valid.

To further see why the invisible devices fail under the governing equation including Equation (Equation 15) when h/L takes 0.1, we plot their simulated velocity distributions in Figure 3(a1,b1,c1). For comparison, we also plot the ideal velocity distribution in Figure 3(a2,b2,c2) according to the velocity transformation rule [Equation ([Disp-formula FD3b-materials-16-00376])]. The ideal distributions themselves demonstrate rigorous hydrodynamic cloaking, concentrating and rotating, while the simulated distributions are significantly different from the former. It is worth noting that there are distinct boundary layers in all subplots. When h/L is not very small, the drag effect from the side walls are obvious. This also shows that when the viscous stress term plays a significant role, Equations (Equation 14) and (Equation 15) can no longer be approximately considered to be form-invariant. At this time, it is unreasonable to exclude the viscous stress term such as Equation (Equation 6). This is consistent with our conclusion that the Stokes equation and the NS equations do not satisfy the form requirement of TO [14]. We have seen one approach to get the devices working again is using the real governing equations of Hele–Shaw flows, which also changes the temperature distribution in the virtual space [Figure 2(b4)]. Another approach preserving the virtual space in Figure 2(b3) is directly putting the ideal velocity distribution into the heat transfer equation [Equation ([Disp-formula FD2a-materials-16-00376])]. The new temperature distributions are illustrated in Figure 3(a3,b3,c3). In addition, we calculate the average temperature deviation in the background [Table 1]. It measures the invisibility effect by spatially averaging the absolute value of the difference between the temperature distribution outside the devices [Figure 2(a3,c3,d3) and Figure 3(a3,b3,c3)] and the corresponding distribution in Figure 2(b3). For all three devices, the average temperature deviation is substantially reduced or even negligible when the velocity is restored to the ideal distribution. We can conclude that the failure of Equation ([Disp-formula FD3b-materials-16-00376]) results in the bad performances of transformation media.

## 4. Multilayered Structures for Transformation Media

As we know, it is very difficult to fabricate anisotropic and inhomogeneous viscosity in experiments, let alone obtain the desired thermal conductivity tensor at the same time. In addition, the cloak requires divergent or zero conductivity and viscosity at its inner edge. In conductive thermal metamaterials, multilayered structures using alternating arrangements of two isotropic homogeneous materials have been used to mimic the extremely demanding properties of transformation media [28]. These structures have also been applied to conduction–radiation systems [27]. Here, we design the similar multilayered structures in thermal convection, a coupled dual-physics scenario.

First, we give the schematic diagram of the cloak in Figure 4(a1). We still use the same background material in the virtual space, and the two materials that make up the cloak are called Material A (in black) and Material B (in white). The cloak consists of 20 concentric annuli in the material order of `A-B-A-B’. Similar structures have been used in the electromagnetic cloak [29]. For a conductive thermal cloak [28], the thermal conductivity of Material A (denoted by $\kappa_{A}$) and that of Material B (denoted by $\kappa_{B}$) satisfy $\kappa_{A}\kappa_{B}=\kappa^{2}$. Similarly, for a hydrodynamic cloak in Hele–Shaw flows, since the pressure satisfies Equation (Equation 11) like the temperature in Fourier’s law, the viscosity of Material A (denoted by $\mu_{A}$) and its counterpart of Material B (denoted by $\mu_{B}$) satisfy $\mu_{A}\mu_{B}=\mu^{2}$. In addition, by analogy with our analysis in realizing a bilayer convective cloak [30], the thermal conductivity and viscosity should have a relationship
(19)$$\kappa_{A}\mu_{A}=\kappa_{B}\mu_{B}$$
to make a convective cloak. This condition comes from the requirement for the synchronous transformation of the thermal and the flow fields. For the cloak, we take h/L=0.001, $\kappa_{A}/\kappa =\mu /\mu_{A}=8$ and $\kappa_{B}/\kappa =\mu /\mu_{B}=\frac{1}{8}$. It can be noticed that the innermost annulus of the cloak is made of Material B, which has a lower thermal conductivity and a higher viscosity. The simulation results are shown in Figure 4(b1,c1). The applied thermal bias, pressure difference and geometric sizes are the same as in the previous simulations. We can see the solid obstacle fails to produce a significant change in the isotherms and isobars in the background region, so this multilayered cloak does work.

In addition, we give the multilayered structures of a concentrator and a rotator in Figure 4(a2,a3), respectively. The concentrator is made up of 90 layers. This rotationally symmetric fan-like structure has also successfully made a magnetic concentrator [31]. Different from a perfect transformation concentrator, the concentrating effect should be influenced by $R_{1}$ and $R_{2}$, and the parameter $R_{m}$ in Equation (Equation 17) does not appear here. The rotator also consists of 90 layers. The helical edge of one layer can be described by the parametric equation, writing $x=R_{1}\exp\left(s\right)\cos\left(ks\right)$ and $y=R_{1}\exp\left(s\right)\sin\left(ks\right)$ ($0\leq s\leq \ln\frac{R_{2}}{R_{1}}$) [28,32]. Here, the rotating effect is determined by the value of $k=-\theta_{0}/\ln\left(R_{2}/R_{1}\right)$ [32], and we still take $\theta_{0}=-\pi /3$. The thermal invisibility of our designs, as well as their hydrodynamic invisibility, can be confirmed by Figure 4(b2,b3,c2,c3). The isotherms inside the concentrator are denser compared with those in Figure 2(b1), showing the effect of magnifying the temperature gradient approximately up to 1.26 times. On the other hand, the isotherms inside the rotator are rotated about −2π/9. The difference between the two materials and the number of layers can also affect the performance of these devices. When more layers and two materials that are more different are used, the amplification ratio of the concentrator (or the bending angle of the rotator) can be closer to $R_{2}/R_{1}$ (or $\theta_{0}$) [31,32].

## 5. Conclusions

In summary, we prove that transformation media inspired by TO can work in Hele–Shaw flows. As model applications, we check the performance of three invisible devices including a thermal cloak, a thermal concentrator and a thermal rotator by numerical simulations. We also design corresponding multilayered structures to realize the similar functions not only for heat transfer, but also for fluid motion. However, our numerical results also show the failure of manipulating heat transfer by transformation media in thermal convection when it occurs in general creeping flows. This is due to the fact that the Stokes equation changes its form under curvilinear coordinate transformations, and the transformed velocity required by the conduction–advection equation cannot be achieved. The thermal convection described by the more general NS equations cannot be strictly form-invariant under curvilinear coordinate transformations either, although they do have some other symmetries such as space-translations, rotations and parity [33] which will not change the material parameters. Our study paves the way for future research in designing convective transformation media in appropriate models. For theoretical rigor, we use a 2D Hele–Shaw model. For more realistic 3D Hele–Shaw cells, we should also consider the 2D transformations on the *x*–*y* plane and creeping flows to obtain accurate manipulation functions [14]. Potential experimental suggestions include modulating the liquid depth or porosity to mimic the required viscosity distribution [14,34] and achieving the thermal conductivity distribution through solid–liquid hybrid metamaterials [30]. The devices we designed can also be extended to the transient regime, such as periodic flows or thermal waves [35], to realize the regulation of wave-like heat transport.

## Figures and Tables

**Figure 1 materials-16-00376-f001:**
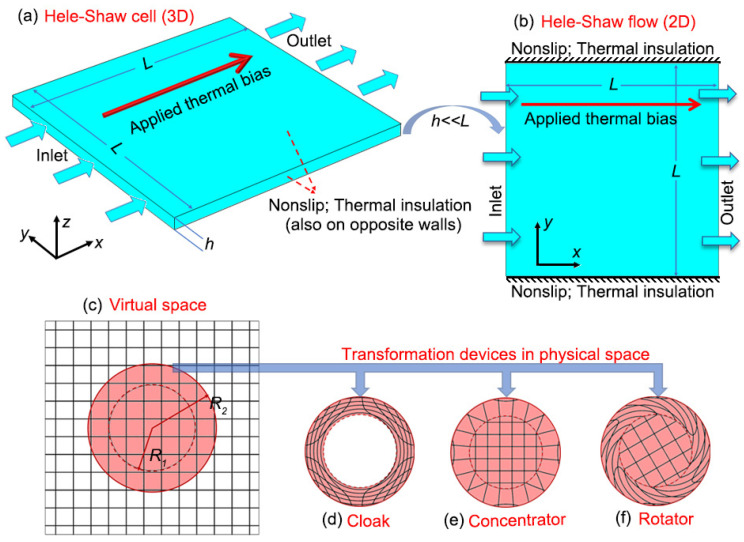
(**a**) Geometry of a Hele–Shaw cell (indicated by transparent color). The dark-colored areas inside represent fluid domain with a depth denoted by *h*. The length and width of the fluid in the horizontal plane are both *L*. The two sides parallel to the *y*-*z* plane are the inlet and outlet of the fluid. The other four sides are solid walls that are insulated and non-slip. A thermal bias is applied along the *x*-axis; (**b**) schematic of a 2D Hele–Shaw flow. It has the same size as the top view of the Hele–Shaw cell. The inlet and outlet are along the direction of the *y*-axis. The boundaries along the direction of the *x*-axis are insulated and non-slip; (**c**) the 2D virtual space for designing transformation media. The red disk undergoes non-trivial transformations to become functional devices. The region outside it undergoes an identity transformation; (**d**,**e**,**f**) are the physical space for the cloak, concentrator and rotator, respectively. The distorted meshes correspond to their respective transformations and functions.

**Figure 2 materials-16-00376-f002:**
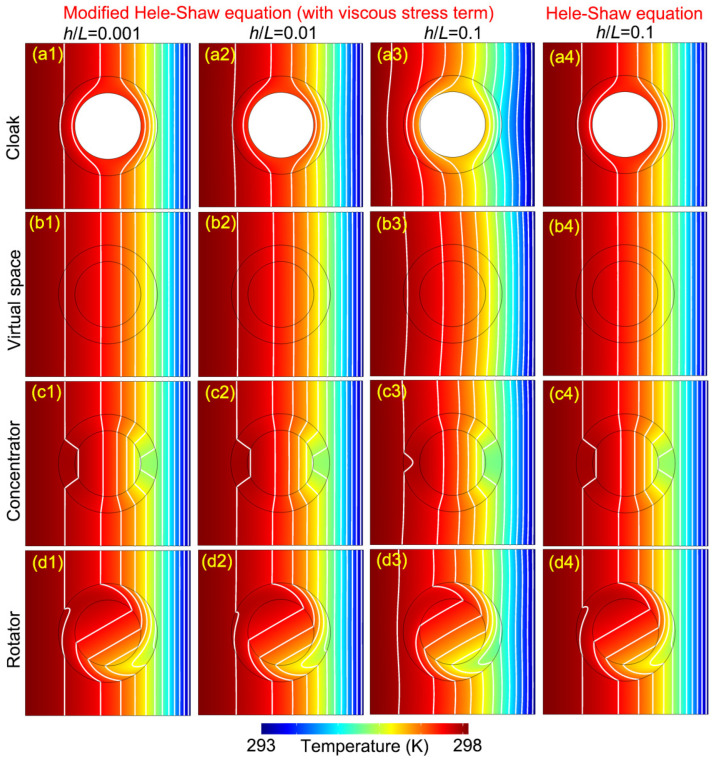
Simulation results of temperature distributions. The white lines are isotherms: (**a1**–**a3**) show the temperature distributions for the cloak when h/L is 0.001, 0.01 and 0.1, respectively. They use Equations (Equation 14) and (Equation 15) in governing equations; (**a4**) shows the temperature distribution for the cloak at h/L=0.1. It represents the result based on the strict Hele–Shaw flow model using Equations (Equation 10) and (Equation 12); (**b1**–**b4**,**c1**–**c4**,**d1**–**d4**) are the corresponding temperature distributions of the virtual space, the concentrator and the rotator, respectively.

**Figure 3 materials-16-00376-f003:**
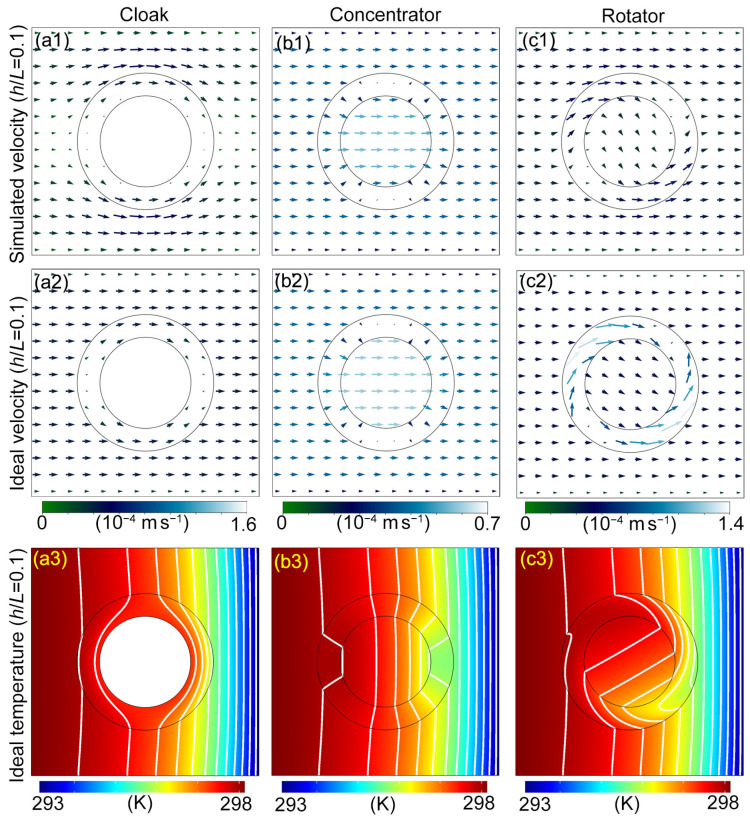
Velocity distribution for invisible devices when h/L=0.1. The arrows indicate the direction of the velocity at their centers. The speeds are related to the lengths and colors of the arrows: (**a1**) is the simulated velocity fields with regard to Figure 2(a3) (the cloak); (**a2**) is the ideal velocity calculated from the transformation rule [Equation ([Disp-formula FD3b-materials-16-00376])]; (**a3**) is the simulated temperature distribution for the cloak under ideal velocity fields when h/L=0.1; (**b1**–**b3**) or (**c1**–**c3**) are the same as (**a1**–**a3**) but for the concentrator or the rotator.

**Figure 4 materials-16-00376-f004:**
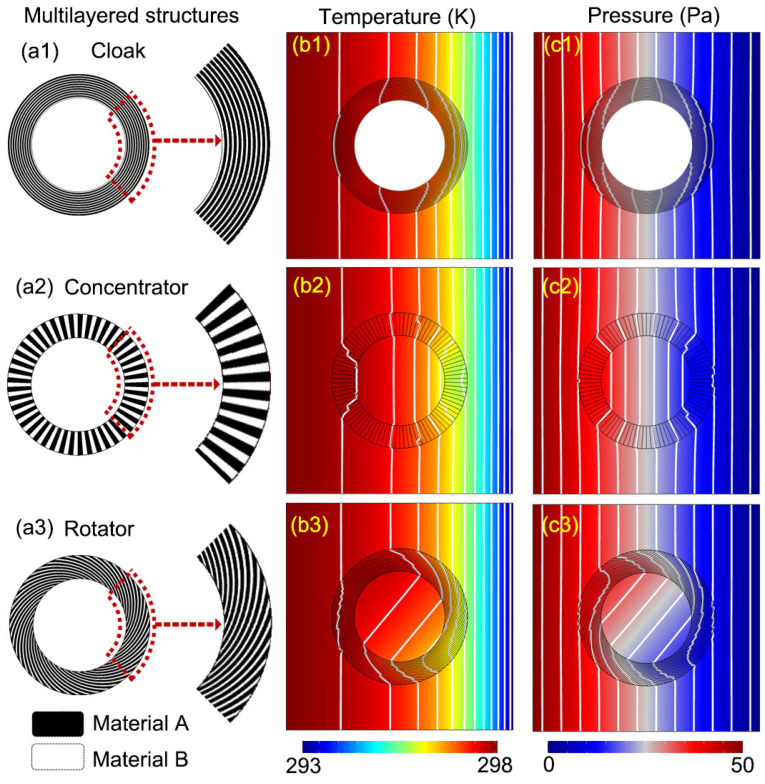
Multilayered structures for convective metamaterials in a Hele–Shaw cell: (**a1**–**a3**) show the structures of the cloak, the concentrator and the rotator made of two alternating materials (Materials A in black and Material B in white), respectively; (**b1**–**b3**) give the simulation results of the temperature distributions for these three devices; (**c1**–**c3**) give the corresponding pressure distributions.

**Table 1 materials-16-00376-t001:** Average temperature deviation in the background when h/L=0.1.

Velocity Type	Cloak	Concentrator	Rotator
Simulated	2.36 K	0.0428 K	0.913 K
Ideal	0.0133 K	$3.75\times 10^{-5}$ K	$2.50\times 10^{-3}$ K

## Data Availability

The data presented in this study are available in this article.

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
