# Peer review of "On Transformation Form-Invariance in Thermal Convection"

_materials, 2022, doi:10.3390/ma16010376_

Round 1

Reviewer 1 Report

The paper "On transformation form-invariance in thermal convection" by Gaole Dai , Jun Wang is well writte and of interest for scientists and engineer working in the field of convection (in particular, in porous media, Hele-Shaw cells and other). In my opinion, the paper deserves to be published in the Journal "Materials", Section "Materials Physics", Special Issue "Thermal Metamaterials and Thermal Functional Devices" after addressiung the following minor issues.  

Discussing the symmetries of the equations of hydrodynamics, I would suggest to cite the book by U.Frisch "Turbulence: The Legacy of A. N. Kolmogorov", CUP, 1996.

It is not clear if the considered 2D Stokes flows are stable to perturbations in the third direction. If not, they are not feasible. PLease, comment on this.

Also, it is not clear if the advective effect (i.e. nonlinear term in the NS equation). in a realistic statment of the problem, can introduce perturbations making the flow unstable. 

Athors do not discuss futute plans at all. It would be interesting if the proposed techniques can be extended to 3D flows. 

Also, since the equations studied are stationary, it would be interesting if the presented analysis can be extended to, at least, a time periodic flows.  

Round 2

Reviewer 2 Report

The authors have addresed all the comments made in the previous review. The article can be recommended for publication.